# Cortical excitability signatures for the degree of sleepiness in human

Chin-Hsuan Chia[1†], Xin-Wei Tang[1†], Yue Cao[1†], Hua-Teng Cao[2,3], Wei Zhang[4], Jun-Fa Wu[1], Yu-Lian Zhu[1], Ying Chen[1], Yi Lin[1], Yi Wu[1], Zhe Zhang[2,3]*, Ti-Fei Yuan[5,6,7]*, Rui-Ping Hu[1]*

[1]Department of Rehabilitation Medicine, Huashan Hospital, Fudan University, Shanghai, China; [2]Institute of Neuroscience, State Key Laboratory of Neuroscience, Center for Excellence in Brain Science and Intelligence Technology, Chinese Academy of Sciences, Shanghai, China; [3]Shanghai Center for Brain Science and Brain-Inspired Intelligence Technology, Shanghai, China; [4]Institute of Brain Science, Fudan University, Shanghai, China; [5]Shanghai Key Laboratory of Psychotic Disorders, Shanghai Mental HealthCenter, Shanghai Jiaotong University School of Medicine, Shanghai, China; [6]Co-innovation Center of Neuroregeneration, Nantong University, Nantong, China; [7]Translational Research Institute of Brain and Brain-Like Intelligence, Shanghai Fourth People's Hospital Affiliated to Tongji University School of Medicine, Shanghai, China

*For correspondence:
zhezhang@ion.ac.cn (ZZ);
ytf0707@126.com (T-FY);
rphu79@163.com (R-PH)

†These authors contributed equally to this work

Competing interest: The authors declare that no competing interests exist.

**Abstract** Sleep is essential in maintaining physiological homeostasis in the brain. While the underlying mechanism is not fully understood, a 'synaptic homeostasis' theory has been proposed that synapses continue to strengthen during awake and undergo downscaling during sleep. This theory predicts that brain excitability increases with sleepiness. Here, we collected transcranial magnetic stimulation measurements in 38 subjects in a 34 hr program and decoded the relationship between cortical excitability and self-report sleepiness using advanced statistical methods. By utilizing a combination of partial least squares regression and mixed-effect models, we identified a robust pattern of excitability changes, which can quantitatively predict the degree of sleepiness. Moreover, we found that synaptic strengthen occurred in both excitatory and inhibitory connections after sleep deprivation. In sum, our study provides supportive evidence for the synaptic homeostasis theory in human sleep and clarifies the process of synaptic strength modulation during sleepiness.

## Introduction

During sleep, brains undergo profound neurophysiological changes that restore the decline in cognitive functions associated with sleepiness (*Harrison and Horne, 2000*; *Tononi and Cirelli, 2006*). While this homeostatic process provides an important opportunity in studying the modulation of cognitive states, the key features of neural circuits underlying wakefulness, sleepiness, and sleep remain to be poorly understood. A synaptic homeostasis theory has been proposed to describe the biophysical change of neural circuits during sleep: wakefulness associates with strengthening of the synaptic connection, while sleep initiates synaptic weight downscaling and facilitates homeostasis (*Tononi and Cirelli, 2006*; *Tononi and Cirelli, 2003*). This theory reasons that constant learning and memory activities during awake lead to synaptic potentiation, thus prolonged awake period causes hyperactivity in the neural circuit, enhancing the noise among neural communications and disrupting cognitive functions (*Tononi and Cirelli, 2003*). It follows that the sleep pressure may be correlated to the degree of this hyperactivity. As increasing noise in the circuits adds barriers for information processing, the organism inevitably falls in sleep to restore the synaptic balance. In a prolonged sleep-deprived state

(>24 hr), this hyperactivation may be partially restored, together with the subjective feeling of sleepiness, by circadian modulation (*Borbély et al., 2016*; *Frank and Cantera, 2014*).

In agreement with this synaptic homeostasis theory, biochemical markers of synaptic potentiation increase with prolonged wakefulness (*Cirelli and Tononi, 2000*; *Tononi and Cirelli, 2001*; *Silva, 2003*). In addition, resting-state electroencephalogram (EEG) revealed a global increase in theta band power (4–8 Hz) with increased sleepiness (*Cajochen et al., 1995*; *Aeschbach et al., 1997*; *Vyazovskiy and Tobler, 2005*), suggesting sleep-wake state exerts a robust modulation on the neural circuits. However, experiments attempting to pinpoint this modulation by directly measuring brain excitability have yielded less consistent results (*Huber et al., 2013*; *De Gennaro et al., 2007*; *Manganotti et al., 2001*). This is at least partially due to the technical challenges in measuring brain excitability, commonly via transcranial magnetic stimulation (TMS). TMS induces noninvasive activation of local brain region by applying a transient, alternating magnetic field through the skull, the effect of which can be readout through EEG or downstream motor output. Brain excitability can be measured by determining the minimal power required to achieve reliable output or the amplitude of the output given a standardized stimulation (*Pascual-Leone et al., 1994*). In addition, paired TMS pulses can be applied consecutively within a short temporal interval (e.g., several milliseconds) to induce certain facilitatory or inhibitory effect (*Stefan et al., 2000*). The nature of these paired-pulse effects has been studied extensively, and some of them have been attributed to the function of certain synaptic receptors (*Van den Bos et al., 2018*; *Daskalakis et al., 2002*). While TMS allows noninvasive assessment of cortical excitability in humans, it suffers from relatively high degree of inter-subject variability. Previous studies taking group averages of TMS measurements reported either hyper- (*Huber et al., 2013*) or hypo- (*De Gennaro et al., 2007*) excitability associated with sleep deprivation, with inconsistent effects of pair-pulse stimulation results (*Manganotti et al., 2001*; *Kreuzer et al., 2011*; *Kuhn et al., 2016*; *Chellappa et al., 2016*).

In this study, we attempted to circumvent the above-mentioned person-to-person variability by applying more advanced statistical methods to analyze the relationship between cortical excitability measured by TMS and self-report sleepiness. We collected TMS measurements in 38 healthy subjects in a continuous 34 hr study program. By utilizing a combination of partial least squares (PLS) regression and mixed-effect models, we identified a robust pattern of cortical excitability change that quantitatively predicts the degree of sleepiness in two separate subgroups of subjects. Interestingly, we found that both facilitatory and inhibitory changes in pair-pulse TMS enhance with sleepiness, while the overall excitability moderately decreases. While our results strongly support the synaptic homeostasis theory that synaptic connection strengthens overtime during wakefulness, we showed that the strengthening occurs in both excitatory and inhibitory synapses. Such changes may contribute to the overall maintenance, and subtle decline, of cognitive functions during prolonged wakefulness (*Rubin et al., 2017*). In sum, our study revealed strengthening of both excitatory and inhibitory synaptic transmission with sleepiness, clarifying and supporting a role of homeostatic modulation of synaptic strength by sleep.

## Results

### Sleep deprivation program

The awake period started after subjects arrived at the hospital and spent the previous night. During the 34 hr of study, the subjects were kept awake with two researchers and took eight times of Stanford Sleepiness Scales (SSS) evaluations and TMS assessments, and three times of EEG measurements (*Figure 1A and B*, *Figure 1—figure supplement 1*). Sleepiness showed an increased after 16 hr (group average SSS scores increased from 2.75 to 4.5), and then a slight decrease after 24 hr (from 4.5 to 3.25) (*Figure 1C*, *Figure 1—figure supplement 2*). This decrease in sleepiness agrees with previous sleep deprivation studies (*Manganotti et al., 2001*), likely due to circadian modulations. Furthermore, consistent with previous reports (*Cajochen et al., 1995*), EEG showed a stereotypical increase in the power of theta band (4–8 Hz), and a decrease in the alpha band (8–12 Hz) (*Figure 1D and E*, *Figure 1—figure supplement 2*). While all brain regions showed consistent global trends, we observed more prominent changes in the occipital areas, consistent with previous studies indicating these areas showed more prominent changes around sleep onsets (*Gorgoni et al., 2019*; *Finelli et al., 2001*). We did not observe an increase in the alpha and beta band in frontal regions, a

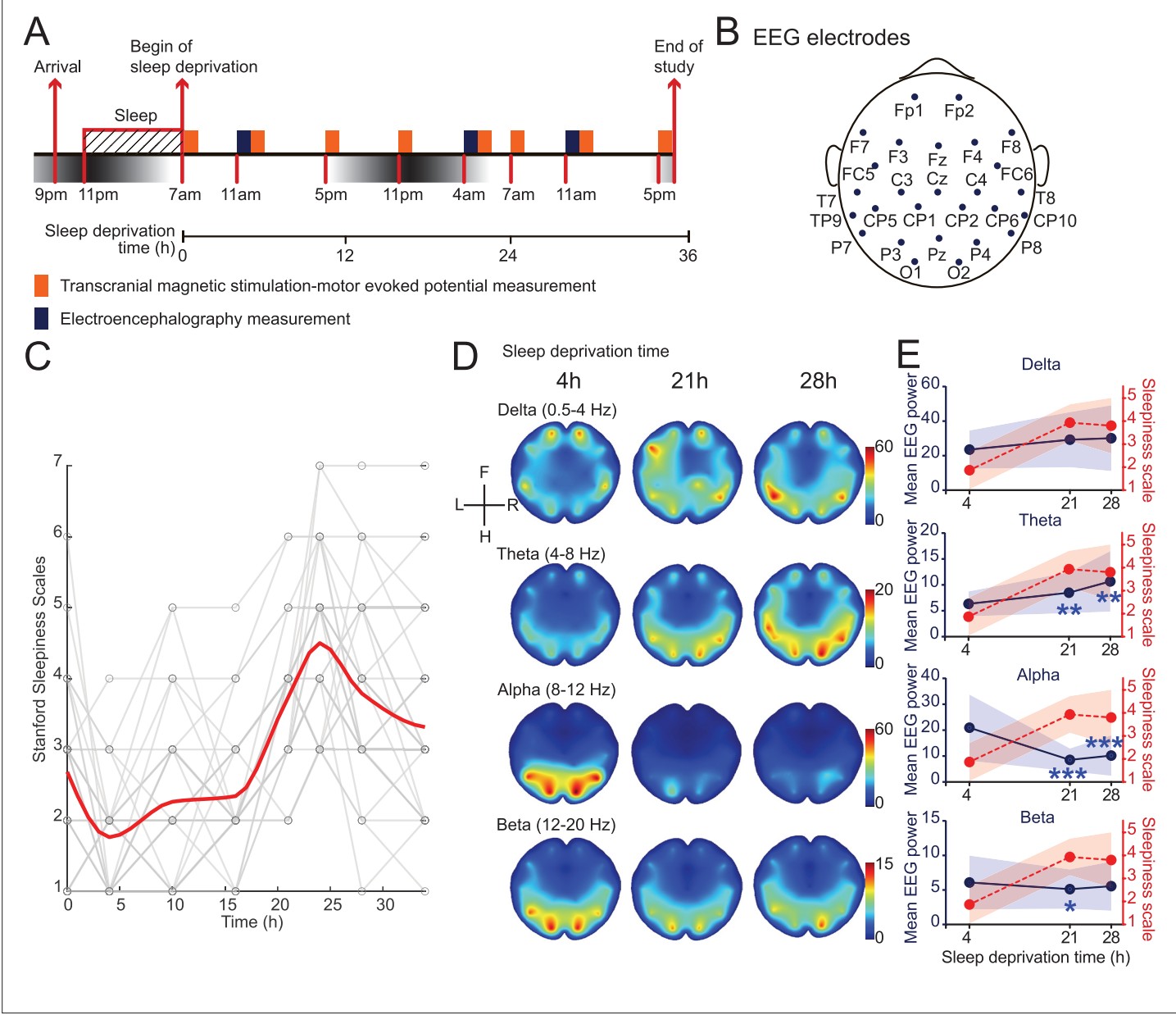

**Figure 1.** Induction and validation of sleep deprivation in current study. (**A**) Schematics of study design. All subjects received eight sessions of transcranial magnetic stimulation (TMS) measurements, but only 15 subjects received electroencephalogram (EEG) measurements (***Figure 1—figure supplement 1***). (**B**) Locations of EEG electrodes used in the study. Positionings are labeled following the extended international 10–20 system. (**C**) Measurements of self-report sleepiness using Stanford Sleepiness Scales (SSS). All individuals received eight measurements. Individuals' scores were showed in gray and averaged responses were plotted in red curve (applied spline interpolation for smooth visualization). Statistical comparisons between sessions are reported in ***Figure 1—figure supplement 2***. (**D**) Averaged heatmaps of different frequency bands' powers in EEG data (N = 15). Interpolations between electrodes were calculated using triangulation-based cubic interpolation method. (**E**) Quantifications of frequency bands' powers showed in (**D**) at different timepoints. Average powers among all electrodes were used for plotting. Red dotted plots show averaged SSS (from **C**). Data are represented as mean ± SD (in shaded areas). Powers at 21 hr and 28 hr are compared with the 4 hr ones using paired t-test with Bonferroni's correction for multiple comparisons. Statistically significant points were labeled (*$p<0.05$; **$p<0.01$; ***$p<0.001$).

The online version of this article includes the following figure supplement(s) for figure 1:

**Figure supplement 1.** Demographic information of participants.

**Figure supplement 2.** All raw data of transcranial magnetic stimulation (TMS) measurements.

prominent feature associated with sleep onset (*Finelli et al., 2001*; *Marzano et al., 2013*), suggesting that the subjects did not fall in and out sleep. In sum, the self-report sleepiness and EEG patterns provided quantitative measurements of the degree of sleepiness in our subjects throughout the study.

### Raw TMS measurements showed weak correlations to sleepiness

We performed TMS stimulation on subjects' motor cortices on the left hemispheres and measured motor-evoked potentials (MEPs) in first interosseous dorsal (FDI) muscle on the right hand (*Figure 2A and B*, *Table 1*). On average, the minimal power necessary for eliciting reliable MEPs decreased after the subjects woke up and mildly increased after 21 hr (*Figure 2C*). The delay between TMS and MEP followed a similar pattern (*Figure 2E*). The increase of minimal power and delay time indicates a reduction in excitability with prolonged wakefulness. However, individuals' data often do not agree with the average trend (*Figure 1—figure supplement 2*), and correlations between individuals' sleepiness score and these parameters showed wide ranges of variability (*Figure 2D and F*). The MT (motor response threshold) and MEP latency are influenced by the excitability along the corticospinal tract. In contrast, the pair-pulse protocols reflect characteristics within the cortex. Among four types of pair-pulse protocols (SICF, SICI, ICF, and LICI; *Figure 2G, J, M and P*), SICI, ICF, and LICI showed strengthening trends on average, yet none were statistically significant (*Figure 2H, K, N and Q*), nor any showed consistent correlations to SSS (*Figure 2I, L, O and R*). In summary, raw TMS measurements showed weak correlation with sleepiness on average, with considerable variance among subjects.

### Sleepiness states can be mapped to latent dimensions in TMS measurements

Considering the individual variability in the TMS measurements, we hypothesized that each subject may have certain intrinsic noise pattern and group averages do not separate these noises from underlying correlation between TMS and sleepiness. In order to explore whether TMS measurements encode sleepiness states, we performed a PLS regression using six TMS parameters as predictors and sleepiness scores as responses (15 subjects, *Figure 1—figure supplement 2*). When we mapped the non-sleepy states (SSS = 1 or 2, blue dots) and sleepy states (SSS = 5, 6, or 7, red dots) on the first two PLS dimensions, we found that the first dimension showed significant separation between these two states, and a mild-sleepy state (SSS = 3 or 4) showed an intermediate peak (*Figure 3A*). Interestingly, while the first dimension explained only ~13 % of the total data variance, other PLS dimensions did not show separation of sleepiness states, suggesting that indeed the TMS data contain high variance, and a particular linear combination of the parameters captures the association to the degree of sleepiness. In comparison, similar analysis using the power of different EEG frequency bands from the same subjects showed obvious separation between sleepy states and non-sleepy states, yet the mild sleepiness showed a bimodal distribution, rather than forming an intermediate cluster (*Figure 3B*). Our results indicated that EEG data exhibits a phasic switch to sleepiness, while TMS data showed a correlation to the degree of sleepiness.

In order to examine whether the observed correlations between sleepiness and TMS or EEG measurements are due to a confounding effect of circadian modulation, we conducted three groups of paired analyses for datapoints at 7 a.m. (sleep deprived 0 and 24 hr), 11 a.m. (sleep deprived 4 and 28 hr), and 5 p.m. (sleep deprived 10 and 34 hr). The separation between no sleepy and sleepy states remained primarily consistent, except for the 5 p.m. group with TMS measurements (*Figure 3—figure supplements 1 and 2*). The lack of correlation in the 5 p.m. group could be due to a lack of sufficient variation in the sleepy states in this timepoint. These data indicate that after controlling for the circadian modulation TMS and EEG measurements still correlate sleepiness.

### TMS quantitatively predicts sleepiness

Based on the conclusion above, we further hypothesized that a mixed-effect model of TMS measurements should be able to predict subjects' sleepiness, while tolerating the high degree of inter-subject variability. We established a linear full model using data from 15 subjects with all six TMS measurements to predict sleepiness scores (*Figure 1—figure supplement 2*), without considering any interaction among these measurements, and each subject was given a random effect on intercept. The model showed a statistically significant fit. We then performed backward variable selection based on

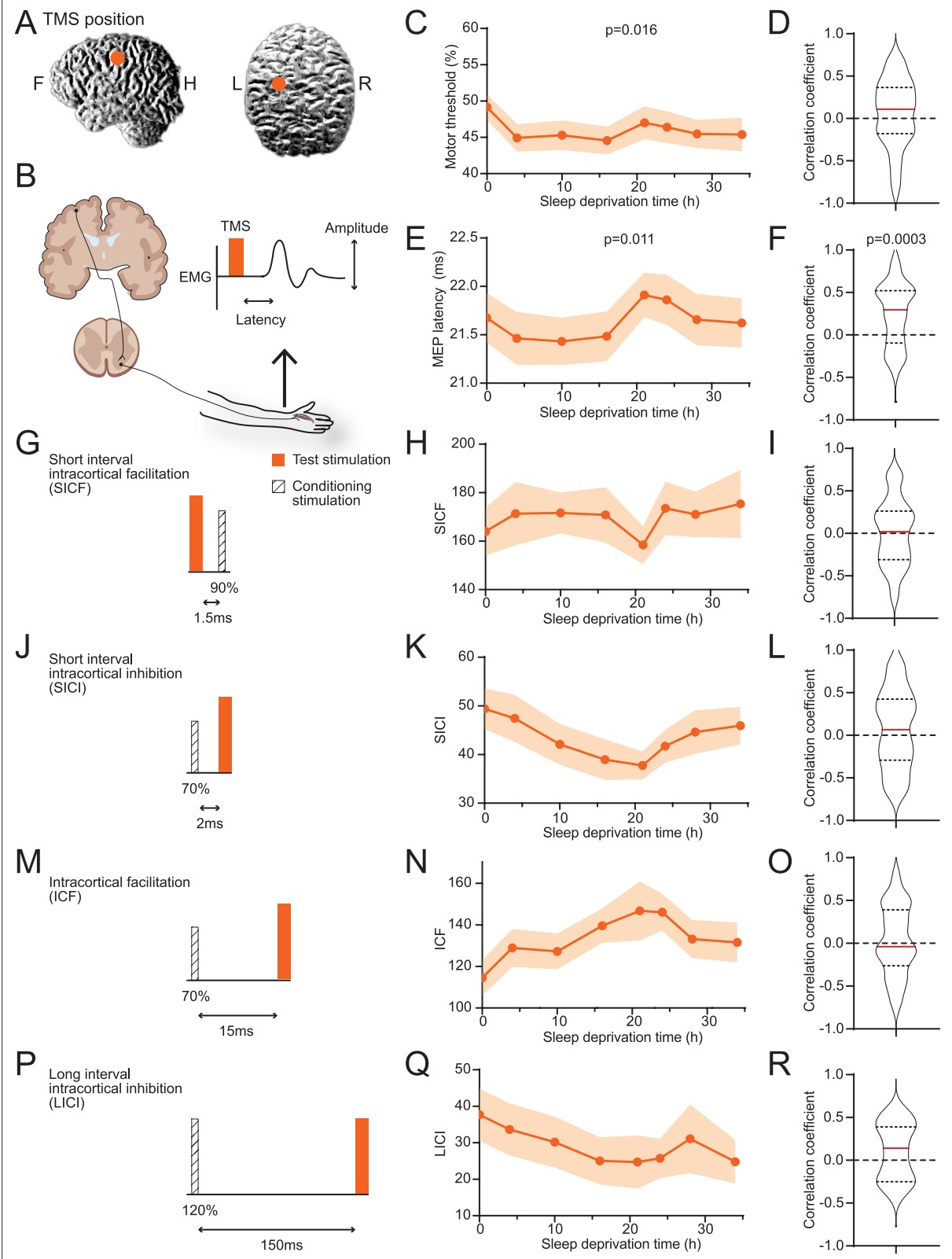

**Figure 2.** Averages of transcranial magnetic stimulation (TMS) measurements show weak correlations to sleepiness. (**A**) Position of TMS targeting. (**B**) Schematics of the TMS study design. The electroencephalogram (EMG) trace showed is for illustrative purpose only. (**C**) Averaged measurements of motor threshold. Data are represented as mean ± SD (in shaded areas). ANOVA test was applied to test the differences among timepoints; p-value is showed when statistically significant. (**D**) Distribution of the correlation coefficients between individuals' sleepiness scores and motor threshold

*Figure 2 continued*

measurements (median in red line, 25% and 75% quantiles in dotted lines). One-sample t-test was used to compare the distribution to 0; p-value is showed when statistically significant. Equivalent to the formats of (**C**) and (**D**), later panels show data for motor-evoked potentials latencies (**E, F**), short interval intracortical facilitations (SICFs) (**H, I**), short interval intracortical inhibitions (SICIs) (**K, L**), intracortical facilitations (ICFs) (**N, O**), and long interval intracortical inhibition (LICI) (**Q, R**). For the four pair-pulse protocols, a schematic is showed for the respective protocol, (**G**) SICF, (**J**) SICI, (**M**) ICF, and (**P**) LICI. For all TMS measurements, N = 38 except for LICI (**Q, R**, N = 15, *Figure 1—figure supplement 1*). Statistical comparisons between sessions are reported in *Figure 1—figure supplement 2*.

F-test of the standard model information criterion (*Figure 3C*, plotting log likelihood as representative), and found that SICF and LICI can be eliminated without reducing the model performance. With the updated model, all four TMS parameters (MT, MEP latency, SICI, and ICF) showed statistically significant coefficients, and the predicted sleepiness showed high correlation to the reported sleepiness (*Figure 3D*). Using the same coefficients on these four TMS parameters, we predicted the other 23 subjects' sleepiness (*Figure 1—figure supplement 2*), allowing a new random effect for each subject. Our prediction showed high degree of correlation to their reported sleepiness (*Figure 3E*), strongly indicating that these coefficients represent a stable pattern in the TMS that robustly correlated with the degree of sleepiness. Using these four parameters and data from all subjects, we performed fivefolded cross-validation of our model (*Colby and Bair, 2013*). Testing group showed performance close to the training group, while shuffled control generated chance-level prediction, further supporting the validity of our model. Similar analysis using TMS measurements from matched SSS states to predict time of the data showed no significant correlation (*Figure 3—figure supplement 2B*), indicating that circadian modulation contributed minimally in the TMS variance in our dataset.

In addition to predicting subjective self-reported sleepiness, we explored the group-level correlation between TMS and EEG measurements using canonical correlation analysis (*Figure 3—figure supplement 3*). We found that the most correlated mode between TMS and EEG is aligned with sleepiness, indicating that sleepiness is the primary covariate linking these two measurements. Finally, we applied the same mixed-effect regression using frequencies power averaged from all EEG electrodes and plotted the variance in the predictions at each degree of reported sleepiness. Consistent with the PLS analysis (*Figure 3A and B*), averaged EEG showed a robust detection of sleepiness, but the gradient was not distinguishable (*Figure 3F*). Together, these analyses showed a robust signature of TMS that uniquely reflects the degree of sleepiness in human.

Furthermore, examining the coefficients in the four TMS parameters revealed that all of them were positive. Therefore, our data indicate that sleepiness is associated with increased MT and MEP latency, both pointing to a decrease in the corticospinal tract excitability. Sleepiness is also associated

**Table 1.** The result of the Stanford Sleepiness Scales (SSS) and transcranial magnetic stimulation (TMS).

| SSS | MT (% MSO) | MEP latency (ms) | SICF | ICF | SICI | LICI | |
|---|---|---|---|---|---|---|---|
| 38 | 38 | 38 | 38 | 38 | 38 | 15 | n |
| 2.68 ± 0.189 | 0.49 ± 0.02 | 21.67 ± 0.26 | 1.64 ± 0.10 | 1.14 ± 0.09 | 0.49 ± 0.04 | 0.39 ± 0.07 | 0 hr |
| 1.76 ± 0.122 | 0.45 ± 0.02 | 21.46 ± 0.27 | 1.71 ± 0.13 | 1.29 ± 0.09 | 0.47 ± 0.05 | 0.35 ± 0.07 | 4 hr |
| 2.26 ± 0.180 | 0.45 ± 0.02 | 2.43 ± 0.25 | 1.72 ± 0.08 | 1.27 ± 0.08 | 0.42 ± 0.04 | 0.31 ± 0.07 | 10 hr |
| 2.34 ± 0.143 | 0.45 ± 0.02 | 21.48 ± 0.26 | 1.71 ± 0.11 | 1.39 ± 0.08 | 0.39 ± 0.04 | 0.26 ± 0.07 | 16 hr |
| 3.74 ± 0.149 | 0.47 ± 0.02 | 21.91 ± 0.23 | 1.58 ± 0.08 | 1.46 ± 0.14 | 0.38 ± 0.03 | 0.24 ± 0.07 | 21 hr |
| 4.50 ± 0.191 | 0.46 ± 0.02 | 21.86 ± 0.26 | 1.73 ± 0.11 | 1.46 ± 0.09 | 0.42 ± 0.03 | 0.25 ± 0.06 | 24 hr |
| 3.79 ± 0.21 | 0.45 ± 0.02 | 21.66 ± 0.27 | 1.71 ± 0.0.9 | 1.33 ± 0.09 | 0.45 ± 0.04 | 0.31 ± 0.09 | 28 hr |
| 3.32 ± 0.223 | 0.45 ± 0.02 | 21.62 ± 0.26 | 1.75 ± 0.14 | 1.31 ± 0.10 | 0.46 ± 0.04 | 0.26 ± 0.07 | 34 hr |

MT: motor response threshold; MSO: maximal stimulation output; MEP: motor-evoked potential; SICF: short interval intracortical facilitation; ICF: intracortical facilitation; SICI: short interval intracortical inhibition; LICI: long interval intracortical inhibition.

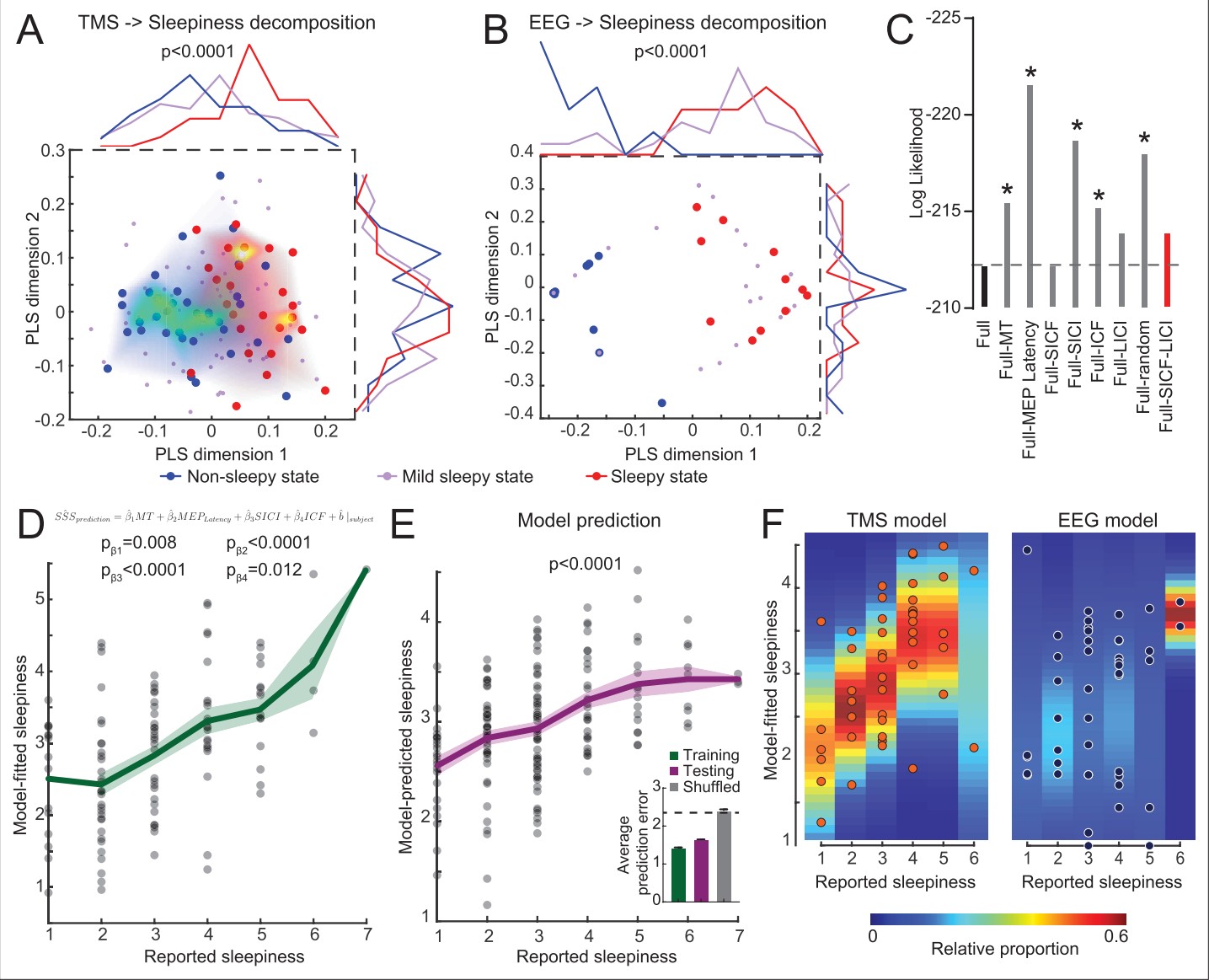

**Figure 3.** Transcranial magnetic stimulation (TMS) measurements quantitatively predict the degree of sleepiness. (**A**) Partial least squares (PLS) regression of TMS measurements to sleepiness scores (15 subjects with eight timepoints each, N = 120 states). The plot shows a scatter plot of each state in the first two PLS dimensions (Stanford Sleepiness Scales [SSS] = 1 or 2 for non-sleepy state, blue dots; SSS = 3 or 4 for mild sleepy state, purple dots; and SSS = 5, 6, or 7 for sleepy state, red dots). The heatmaps show density plots of the blue and red dots. And the distributions along each dimension are plotted on top and right side of the plot. Student's t-tests were used to compare the distributions between sleepy and non-sleepy states, with p-value indicated if statistically significant. (**B**) PLS regression of electroencephalogram (EEG) measurements (15 subjects with three timepoints each, N = 45 states), similar to panel (**A**). (**C**) Log likelihood of different regression models. Increase in absolute values indicates a worse fit. Dash line indicates the level of full model. All models were compared to the full model using F-test of all four information criteria: Akaike Information Criterion (AIC), Bayesian Information Criterion (BIC), log likelihood and deviance; with asterisks showing statistical significance. The red bar indicates the model chosen for later analysis. (**D**) Mixed-effect linear model fitting results from the training set (15 subjects with eight timepoints each, N = 120 states). Green line shows mean ± SEM of the model predictions. p-Values of each coefficient was showed. (**E**) Mixed-effect linear model fitting results from the testing set (23 subjects with eight timepoints each, N = 184 states). All coefficients were kept the same with the training set in (**D**), except allowing a new random effect on intercept for each new subject. p-value was calculated with a F-test of a linear regression between predicted values to reported values to show the slope is significantly different from 0. Inset in (**E**) shows average performance from fivefold cross-validation compared to shuffled control. (**F**) Comparison of mixed-effect linear model fittings between TMS and EEG measurements, with balanced dataset (15 subjects with three timepoints each, N = 45 states for both). Heatmaps were calculated by fitting a normal distribution density function with the predicted values at each reported value bin.

The online version of this article includes the following figure supplement(s) for figure 3:

**Figure supplement 1.** Partial least squares (PLS) regression of electroencephalogram (EEG) measurements to Stanford Sleepiness Scales (SSS).

*Figure 3 continued on next page*

*Figure 3 continued*

**Figure supplement 2.** Controlling for circadian modulation of transcranial magnetic stimulation (TMS) measurements.

**Figure supplement 3.** CCA analysis between transcranial magnetic stimulation (TMS) and electroencephalogram (EEG) measurements.

with enhancement in both facilitatory (ICF) and inhibitory (SICI) pair-pulse effects, suggesting the strengthening of both excitatory and inhibitory synapses during prolonged wakefulness.

## Discussion

The present study reported that a combination of four TMS parameters (rMT, MEP latency, SICI, ICF) could efficiently predict subjective sleepiness. The values of rMT and MEP latency together directly reflect and measure the excitability states of motor cortical region. Enhanced rMT has been reported using similar sleep-deprivation paradigm (*Manganotti et al., 2001*). And increases in cortical excitability were reported using local EEG measurements of TMS-evoked responses (*Huber et al., 2013*). On the other hand, the pair-pulse parameters, SICI and ICF, reflect local changes in the cortex. SICI is believed to be mediated by GABAAR, and ICF is mediated by glutamatergic transmission, potentially through NMDAR (*Van den Bos et al., 2018*). Therefore, our study suggested that both excitatory and inhibitory synapses are strengthening with prolonged wakefulness. Previous studies demonstrated that cortical excitability can be sensitive to many variables including age (*Gaggioni et al., 2019*), attention (*Cardone et al., 2021*), and circadian rhythm (*Chellappa et al., 2016*; *Ly et al., 2016*). Among these, effects of circadian rhythm are challenging to disambiguate since sleepiness correlates to circadian factors (*Figure 1*). Thanks to the prolonged durations of our study, we were able to compare timepoints 24 hr apart (*Figure 3—figure supplement 2*), which were not available to previous studies (*Chellappa et al., 2016*; *Ly et al., 2016*). We found that the correlation between TMS measurements and sleepiness remains significant after controlling for the circadian factors; but TMS measurements do not show correlation to circadian timing within each SSS subgroup. These data strongly suggest that the variance in cortical excitability is primarily correlated with sleepiness in our dataset. However, in other situations with short or no sleep deprivation, it is still likely that circadian factors are correlated with cortical excitability.

This finding agrees with the general theme of the synaptic homeostasis theory of sleep: wakefulness is associated with continuous strengthening of the synapses. Importantly, we provided an additional supplement to this theory that the synaptic strengthening does not only happen in excitatory synapses, but also in inhibitory synapses. Lacking the exact mechanism of these changes, it is possible the observed strengthening in both types of synapses was independently modulated by sleepiness or is due to the circuit response to changes in one type of the synapse in order to maintain the excitation/inhibition balance (E/I balance). Thus, in contrast to the original prediction of the theory that sleepiness leads to overt hyperactivity, we showed that there is a robust E/I balance. Nevertheless, the strengthening of synapses while maintaining balance can still have a detrimental effect on the information processing in the circuits, as demonstrated by multiple studies simulating these effects. Deviating from the optimal range of the synaptic strengths will lead to a reduction in the signal-to-noise ratio in individual neurons (*Rubin et al., 2017*), as well as the suboptimal sparseness and noise correlations in the population-wide coding (*Zhou and Yu, 2018*). Sleep, then, may depress these synaptic weights and restore the circuits to an optimal state (*Kuhn et al., 2016*).

While the above-mentioned modification of the synaptic homeostasis theory does not change the general framework of sleep function, it is worth noting that the difference is not just a pedantic discussion. Our finding predicts that general excitability will not be a reliable marker for sleepiness. In addition, if any intervention were to develop to relief sleep drive, our finding suggests that prolonged synaptic suppression will not serve a restorative function. These predictions differ from the ones posited from the original synaptic homeostasis theory and will have important implications in the future research on the monitoring and modulation of sleep states.

Finally, we reported that a linear combination of TMS parameters can predict the gradual extent for the degree of sleepiness, while using the averaged EEG data only categorically separated non-sleepy and sleepy states. Although EEG was only recorded on three timepoints, limiting the TMS model using the same amount of data still produced a gradient prediction for the degree of sleepiness (*Figure 3*), suggesting that there is likely to be some intrinsic different modes of modulation to the

sources of EEG and TMS measurements by sleepiness. As many frequency bands are empirically and potentially mechanistically linked to neural network function (*Başar et al., 2001*; *Klimesch, 1999*), the binary switch of EEG states may reflect a robust scheme of neural computation up to a high degree of sleepiness, and the change in the EEG state is potentially more predictive to the cognitive function decline related to sleepiness, compared to changes in the TMS measurements. Yet the synaptic changes revealed by TMS measurements may represent a mechanism underlying the EEG state switch as the synaptic strengths could impose biophysical boundaries and stable-state landscapes that force the switch in rhythmic activities states. On the other hand, a previous EEG and intracranial recording study showed that sleep onset is associated with a strong theta rhythm in the occipital regions and an increase in alpha and beta rhythm in the frontal regions (*Marzano et al., 2013*), suggesting that sleep deprivation may lead to different changes in rhythmic activities and potentially excitabilities in different brain regions. It remains to be seen whether our current finding represents a general feature across the brain.

Although our sleep deprivation paradigm exceeded the duration of many previous studies, one limitation of the current study is the limited datapoints of level 6–7 sleepiness. While we observed robust correlation with SSS levels of 1–5, the statistical power for very sleepy states is lower. This may partially explain the lack of correlation in one of the subsets in our results (*Figure 3—figure supplement 2*).

In conclusion, our study showed that the strengthening of both excitatory and inhibitory synaptic connections in the cortex can quantitatively predict the degree of sleepiness. This finding indicates a modification of the synaptic homeostasis theory of sleep and furthers our understanding of how sleepiness state modulates brain functions.

## Materials and methods

### Participants

Thirty-eight healthy participants (21 males, 17 females, age: 22.46 ± 0.29 years, age range: 20–27 years, *Figure 1—figure supplement 1*) with no history of sleep disorders (Pittsburgh Sleep Quality Index; *Buysse et al., 1989*), head trauma, psychiatric conditions, or any other chronic disease were recruited for present study. The subjects were free of medication and tobacco use. All participants wrote informed consent before the experiment; the study has been approved by the Ethics Committee of Huashan Hospital (2017-410).

### Study flow

The subjects were asked to stay in their regular sleep pattern at least 1 week before the experiment, and the experiment would postpone if the participants have a cold or any other uncomfortable situation. The subjects arrived at the laboratory 1 day before the experiment for preparation and to adapt to the environment. They slept in the separate and quiet room for one night. They were asked not to use the alarm and are wakened by the researchers at 7 a.m. of the second day.

The study lasts for 34 hr, during which the subjects received eight times for SSS evaluation and TMS assessments, as well as three times resting-state EEG measurements (*Figure 1*). SSS was used in our study due to its wide usage in research, previous validation against objective measurements, and brevity (*Hungs, 2012*). During the study period, the subjects were not allowed engaging in vigorous exercise or taking coffee or tea, and were maintained awake by two independent researchers.

### EEG recording and data processing

The EEG signal was recorded by EEG system of BrainAmp MR32, BrainProducts, using a 32-channel EEG cap. A specific electrode was used as the reference channel placed between Cz and Fz. Reference would then be transferred offline into overall average reference. Subjects were seated in a comfortable chair with a computer screen in front of them showing instructions when recording the resting state EEG. Subjects were asked to focus on a cross symbol on the screen for 5 min to record their open-eye EEG and followed by another 5 min of close-eye EEG epoch and then repeated for totally 20 min EEG recording with two episodes of open and close eye EEG, respectively. The EEG was acquired at 4, 21, and 28 hr from awake.

**Table 2.** The setting of the ppTMS.

|  | SICI | LICI | SICF | ICF |
|---|---|---|---|---|
| CS (%MEP) | 70 | 120 | 90 | 70 |
| TS (%MEP) | Average amplitude reached at 400–1000 µV | | | |
| ISI | 2 | 150 | 1.5 | 15 |

ppTMS: paired-pulses transcranial magnetic stimulation; SICI: short interval intracortical inhibition; LICI: long interval intracortical inhibition; SICF: short interval intracortical facilitation; ICF: intracortical facilitation; MEP: motor-evoked potential; CS: conditioning stimulation; TS: test stimulation: ISI: interstimulus interval.

The collected EEG signals were processed offline using MATLAB (The MathWorks Inc, Natick, MA). Primary processing included eye-blink correction, artifact rejection, segmentation, and band-pass filter. The eye-blinks were removed from the EEG signal in other channels to eliminate the effects of electro-ocular artifacts. High-frequency artifacts, such as muscle activity, and high-amplitude slow wave were rejected based on the automatic removal algorithm with an initial threshold of 70 µV, which would continuously increase 5 µV at a time if more than 20 % of the data was rejected until the threshold reached 150 µV. According to the study protocol, the EEG signal was then segmented by 1 min and following band filter of 0.5 –49 Hz as beta wave was often interested in sleep deprivation studies.

The EEG signals were decomposed into the delta (1–4 Hz), theta (4–8 Hz), alpha (8–12 Hz), and low-beta (12–20 Hz) frequency components, spectral power ($\mu V^2$) of which was analyzed separately via fast Fourier transform (FFT) for overall average at the three timepoints, respectively.

## TMS procedures

The subjects were seated on a comfortable chair in a silent environment. TMS studied was conducted with OSF–priming TMS (YRD CCY-IA, Yiruide Co., Wuhan, China) connected with a 70 mm figure-of-8 coil (CCY-I TMS instrument, Yiruide Co.). The motor hotspot for FDI muscle on the contralateral hemisphere of primary motor cortex (M1) was defined with maximum MEP value. The MT is defined as the lowest density that can be evoked at least 5 MEP, with amplitude more than 50 µV in 10 times of successive stimulation with the relaxed recording muscle (*Rossini et al., 2015*). Then, the intensity of the test stimulation (TS) was determined when the average amplitude reached at 400–1000 µV. The TS was given 10 times continuously, and their latency and amplitude were recorded.

The paired-pulses TMS (ppTMS) included short interval intracortical inhibition (SICI), long interval intracortical inhibition (LICI), short interval intracortical facilitation (SICF), and intracortical facilitation (ICF). The ppTMS also included three controllable parameters, TS, conditioning stimulation (CS), and interstimulus intervals (ISI). The setting of ppTMS in this study is shown in *Table 2* (*Rossini et al., 2015*; *Lazzaro and Ziemann, 2013*). Every index was repeated 10 times, and the single pulse of the TS was inserted between the pulses to guarantee the accurate stimulation site and the angle of the coil. The pause was set within 5–7 s between the pulses to make sure the excitability would not have been interfered by the last stimulation. The format of the index of the ppTMS is shown below. The 10 times stimulation of each index were averaged.

$$\text{ppTMS}\left(\%\text{MEP}\right) = \frac{amplitude\ of\ ppTMS}{amplitude\ of\ TS} \times 100\%$$

## Statistical analysis

The data were repeated measurement variables of single group. One-way ANOVA was used for statistics, and the sample size was calculated based on the data from previous studies. TMS and SSS were repeated seven times and EEG three times. PASS 11 was used to calculate the sample size of 34 (power = 0.9, $\alpha$ = 0.05 [double-sided]). If the lost rate was assumed to be 10%, then the sample size was 34/0.9 = 38 cases. In the actual study, 40 cases were included and lost 2 cases.

All statistical analyses were performed with GraphPad Prism (GraphPad Software LLC, USA). Paired t-tests were used in EEG dataset (*Figure 1*). One-way ANOVA was applied in the result of TMS measurements and one-sample t-test for the correlations between TMS and SSS (*Figure 2*). The statistical significance threshold was set as two-tailed, p<0.05. F-tests were used in most of the model-related coefficient comparisons (*Figure 3*). Data are presented as mean ± standard deviation or mean ± standard error of the mean.

Partial least-square regressions were performed with MATLAB, using SSS as responses. In the case of TMS measurements, all six TMS parameters were used as predictors and output was set with five dimensions. In the case of EEG measurements, all four EEG bands were used as predictors and output was set with three dimensions. The correlation between SSS and the predictor scores for each output dimensions was examined, and the first two dimensions were plotted.

Mixed-effect linear regressions were performed with MATLAB, using SSS as responses. All parameters were normalized to a 0–1 scale according to group extrema before used in the model. Pilot explorations adding sex and age as predictor variables show no effect of these variables. Additionally, using all subjects' data, all random effects and residuals follow normal distribution and are not correlated, suggesting compliance to the core assumption of the mixed-effect model. No interaction term was allowed, and the random effect was limited to only the intercept for each subject. Backward model selection was performed based on information criteria compared to the full model. It should be noted that parameters not selected may still change with SSS, but would not provide further information in a prediction model. To apply the fitted model in a new group, we calculated the fixed effects of each new subject using the fitted coefficients and then fit a new model adding only a random intercept for each new subject.

For cross-validation of our model, we adopted the methods described in a previous study (*Colby and Bair, 2013*). In brief, for each iteration, all subjects were randomly assigned into training (23 out of 38) or testing (15 out of 38) group. Within training group, 18 subjects were used to estimate model coefficients. After freezing the fixed-effects coefficients, random effects were estimated for the remaining five subjects and residuals from these five subjects were used as model performance. This within-training group sampling was repeated 500 times and the best model is selected. Using the fixed-effect coefficients from this model, we re-evaluated the random effects from all 23 subjects and calculated the residuals as training group performance. And then we applied this model to the 15 subjects and calculated the random effects and residuals, which were recorded as testing group performance. This concluded one iteration of the cross-validation process. 500 iterations were performed on the dataset, and averaged performance was reported in the results. We conducted the same procedures in randomly shuffled SSS data, and the testing performance in that group is recorded. To estimate chance level, we draw 38 random number from 1 to 7 following a normal distribution as model prediction and calculated the errors. This procedure was repeated 500 times to get an average chance level.

## Acknowledgements

This work was supported by the National Key R&D Program of China (2018YFC2001700), National Natural Science Foundation of China grants (81822017, 31771215, 32071010), Medicine and Engineering Interdisciplinary Research Fund of Shanghai Jiao Tong University (ZH2018ZDA30), the Key Projects of Shanghai Science and Technology on Biomedicine (18411962300), Shanghai Health and Family Planning Commission project (201840225), Shanghai Municipal Key Clinical Specialty (shsl-czdzk02702), the Projects of Shanghai Science and Technology (20412420200), Shanghai Municipal Education Commission - Gaofeng Clinical Medicine Grant Support (20181715), Shanghai Pujiang Program (20PJ1415000), Shanghai Municipal Science and Technology Major Project (2018SHZDZX05), and the State Key Laboratory of Neuroscience.

## Additional information

### Funding

| Funder | Grant reference number | Author |
| --- | --- | --- |
| National Key Research and Development Program of China | 2018YFC2001700 | Yi Wu |

| Funder | Grant reference number | Author |
|---|---|---|
| Medicine and Engineering Interdisciplinary Research Fund of Shanghai Jiao Tong University | ZH2018ZDA30 | Ti-Fei Yuan |
| Natural Science Foundation of China | 81822017 | Ti-Fei Yuan |
| Natural Science Foundation of China | 32071010 | Zhe Zhang |
| Natural Science Foundation of China | 31771215 | Ti-Fei Yuan |
| Shanghai Science and Technology Development Foundation | 18411962300 | Yu-Lian Zhu |
| Shanghai Municipal Commission of Health and Family Planning Foundation | 201840225 | Rui-Ping Hu |
| Key Clinical Research Program of Southwest Hospital | s.shslczdzk02702 | Yi Wu |
| Shanghai Science and Technology Development Foundation | 20412420200 | Yi Wu |
| Shanghai Pujiang Program | 20PJ1415000 | Zhe Zhang |
| Shanghai Municipal Science and Technology Major Project | 2018SHZDZX05 | Zhe Zhang |
| State Key Laboratory of Neuroscience | | Zhe Zhang |
| Shanghai Municipal Education Commission - Gaofeng Clinical Medicine Grant Support | 20181715 | Ti-Fei Yuan |

The funders had no role in study design, data collection and interpretation, or the decision to submit the work for publication.

## Author contributions

Chin-Hsuan Chia, Investigation, Methodology, Project administration, Writing - original draft; Xin-Wei Tang, Investigation, Project administration; Yue Cao, Project administration, Software, Writing - review and editing; Hua-Teng Cao, Data curation, Formal analysis, Writing - review and editing; Wei Zhang, Software, Visualization; Jun-Fa Wu, Resources; Yu-Lian Zhu, Data curation; Ying Chen, Methodology, Software; Yi Lin, Project administration; Yi Wu, Funding acquisition; Zhe Zhang, Data curation, Formal analysis, Software, Writing - original draft, Writing - review and editing; Ti-Fei Yuan, Conceptualization, Supervision; Rui-Ping Hu, Supervision

## Author ORCIDs

Chin-Hsuan Chia (iD) http://orcid.org/0000-0002-3343-9541
Zhe Zhang (iD) http://orcid.org/0000-0002-0899-8077
Ti-Fei Yuan (iD) http://orcid.org/0000-0003-0510-715X
Rui-Ping Hu (iD) http://orcid.org/0000-0002-7626-2061

## Ethics

Human subjects: 1. That informed consent, and consent to publish, was obtained 2. This study was designed as a prospective self-controlled study. The Ethics Committee of Huashan Hospital approved the study (2017-410) and was registered on the Chinese Clinical Trial Registry (ChiCTR1800016771).

Decision letter and Author response
Decision letter https://doi.org/10.7554/eLife.65099.sa1
Author response https://doi.org/10.7554/eLife.65099.sa2

## Additional files

### Supplementary files
- Transparent reporting form
- Source data 1. Raw data collected in this study.

### Data availability
All data generated or analysed during this study are included in the manuscript and supporting files.

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
