## [Decision Letter]

**Acceptance summary:**

This paper is of potential interest to neuroscientists and sleep researchers, as it links measures of cortical excitability to self-reported sleepiness. The dataset is interesting given all the various measures reported. The authors revised the manuscript in depth according to the reviewers' suggestions, with some new control analyses, including the relative independence from circadian factors.

**Decision letter after peer review:**

Thank you for submitting your article "Cortical excitability signatures for the degree of sleepiness in human" for consideration by *eLife*. Your article has been reviewed by 3 peer reviewers, and the evaluation has been overseen by a Reviewing Editor and John Huguenard as the Senior Editor. The following individuals involved in review of your submission have agreed to reveal their identity: Giulia Gaggioni (Reviewer #2).

Based on the reviews, and as detailed below we would like to see a revised version of your manuscript before making a final decision. The reviewers have discussed their reviews with one another, and the Reviewing Editor has drafted this to help you prepare a revised submission.

Essential revisions:

1. The main concern raised by all three reviewers regards the impossibility to disentangle between the contributions of circadian vs. homeostatic factors on the observed effects. The analyses have been performed only considering the homeostatic factor (i.e., the time spent awake). We recommend that you address this concern by the following additional analysis: comparisons between data points collected at the same time-of-day (i.e., controlling the circadian factor): 3 data points for the TMS measures [7 a.m, 11 a.m., and 5 p.m. of the first and second day (before vs. after sleep deprivation)] and 1 data point for EEG measures (11 a.m. of the first and second day).

2. Linking the wakefulness measure to synaptic homeostasis theory may be possible via literature references. It would be important to explain (in the introduction, at least) what synaptic homeostasis theory predicts about a >24h sleep-deprivation intervention including the 'rebound' effect on sleepiness after 24h, since that is what the authors have done.

3. It would also make sense to test the model more rigorously with something like 5-fold cross-validation.

4. Line 352: "Pilot explorations adding sex and age as predictor variables show no effect of these variables." It would be interesting to know if the predictor "session" was also tested.

5. Figures 1.C. and 2.C-E.: if there are any significant differences between sessions, please report it in the figure.

6. Figure 1.D..: could you discuss the higher δ and theta activity observed in occipital areas (instead of fronto-central areas)?

7. Line 128: MT abbreviation without previous definition in the text (but in line 307).

8. Methods / Statistical analysis: specify if the main assumptions required by mixed-effect model where checked and met.

9. Methods / Study flow: give a brief rational why SSS (instead of, for example, KSS, VAS, etc.).

10. An interesting analysis could be to test if the model could also predict the EEG activity (mean or theta), as an objective marker of sleepiness (even though EEG was only recorded 3 times in a subgroup of 15 participants).

11. The study participants do not actually report feeling very sleepy. The key behavioral measure is self-reported sleepiness on the Stanford Sleepiness Scale which is a 7-point scale, with only 6-7 indicating the participant is 'sleepy'. However participants reported level 6-7 sleepiness only at approximately 15 observations (fig3D, grey dots). This is from 38 participants * 8 testing moments i.e., >300 total observations. If participants were only actually sleepy in <5% of observations this could weaken the conclusions about sleep and sleepiness. And the clustering of the observations around the lower points 1-5 on the Sleepiness Scale might reduce confidence in the model fit. This is not necessarily a big issue but it might be sensible to acknowledge it in the discussion.

*Reviewer #1:*

This study investigates in 38 healthy subjects the association between cortical excitability increases, as assessed by transcranial magnetic stimulation (TMS), and sleepiness induced by a 34-hour protocol of sleep deprivation.

Using advanced statistical methods, the study confirms that the cortical excitability changes are associated to the degree of sleepiness. They also report that synaptic strengthen after sleep deprivation occurred in both excitatory and inhibitory connections.

On the whole, the study is essentially replicative. Differently from what stated by the authors ("Previous studies taking group averages of TMS measurements reported either hyper- [10] or hypo- [11] excitability associated with sleep deprivation, with inconsistent effects of pair-pulse stimulation results [12, 17-19]."), the existing studies are mostly coherent. Refs. #11, #12, and 18 substantially confirm the hypothesis. Refs. #12 and #17 are methodologically weak studies providing inconsistent findings, while Ref. #19 highlights the contribution of both circadian-timing and prior sleep-wake history in regulating cortical excitation/inhibition balance.

On the other hand, the most novel aspect of the current study is represented by the use of advanced statistical methods, which allow to control intrinsic individual variability in the TMS measures. The second novel finding of the study is that the synaptic strengthening does not only happen in excitatory synapses, but also in inhibitory synapses.

Taking into consideration the protocol (please see panel A of Figure 1), TMS measures were collected in eight different datapoints [7 a.m., 11 a.m., 5 p.m., and 11 p.m. of the first day (i.e., without any sleep deprivation) and 4 a.m. 7 a.m. 11 a.m. 5 p.m. of the second day (i.e., after sleep deprivation)]. On the other hand, EEG measures were collected at 11 p.m. of the first day and at 4 a.m. and 11 a.m. of the second day.

The study seems not considering that consolidated models of sleep regulation (Borbély AA, Daan S, Wirz-Justice A, Deboer T. The two-process model of sleep regulation: a reappraisal. J Sleep Res. 2016 Apr;25(2):131-43) state that nightly sleep and diurnal sleep pressure are affected and regulated by circadian-timing (i.e., the circadian process) and prior sleep-wake history (i.e., the homeostatic factor). This means that considering the 8 TMS datapoints and the 3 EEG datapoints intrinsecally mix the effects of circadian and homeostatic factors.

According to this very basic point, the current results can't disentangle in any way the contributions of the two factors. The analyses have been performed only considering the homeostatic factor (i.e., the time spent awake).

*Reviewer #2:*

In this study, Zhang and colleagues tracked the time course of sleepiness (self-reported), of the EEG, and of 6 indices of cortical excitability (MT, MEP latency, SICF, SICI, ICF, LICI), measured by TMS and paired-pulse TMS at the motor cortex, during 34 hours of sleep deprivation in 38 subjects (EEG and SICI measured in a subgroup of 15 subjects in 3 out of 8 sessions). Only MT and MEP latency showed a significant change with time spent awake. MEP latency was also significantly associated with self-reported sleepiness. The profile of the 6 raw indices showed high variability between individuals. Thus, the authors decided to apply advanced statistical methods such as partial least squares (PLS) regression and mixed-effect models to further analyse the relationship between cortical excitability measures and self-reported sleepiness. Zhang at al. identified a model based on the combination of 4 indices of cortical excitability (MT, MEP latency, SICI, ICF) that quantitatively predicts the degree of sleepiness in two separate subgroups of subjects. Furthermore, based on the regression coefficients of the model, they found that -while corticospinal excitability decreases- both facilitatory and inhibitory changes enhance with sleepiness. These data from a human in vivo protocol add in an interesting twist to the current frame of the SHY theory. The conclusions of this paper are mostly well supported by data driven from the model. However, the raw data are difficult to decipher. Some aspects of the volunteers' recruitment and final claims need to be clarified and extended.

Strengths

The duration and the number of sessions of the protocol are remarkable, as well as the number of participants. The design of the study is articulated and includes questionnaires, EEG and TMS / ppTMS over the motor cortex. Zhang at al. investigated both cortex excitability (i.e. reactivity) through motor-evoked potentials as well as the excitatory and inhibitory balance directly at the motor cortex. In this paper, cortical excitability measures are linked to self-reported sleepiness during an extended wakefulness, possibly representing the neurophysiological substrates of various aspects of human cognitive performance during sleep deprivation.

The authors applied a combination of PLS and mixed model and showed that self-reported sleepiness can be elegantly predicted by a set of indices of cortical excitability. This model allows a fine-grained prediction compared to the bymodal EEG one. This model recalls and further disentangles the positive association between cortical excitability (TEP amplitude) and subjective sleepiness previously reported during a 29 hrs sleep deprivation protocol under constant routine conditions (Ly, J. Q. M. et al. Circadian regulation of human cortical excitability. Nat. Commun. 7:11828 doi: 10.1038/ncomms11828 (2016)).

Weaknesses

Person-to-person variability of the raw data, resulting in weak correlation of the raw data with self-reported sleepiness. This inter-variability could potentially have been minimised, at least in part, by:

1. Designing the sessions based on time elapsed since awake (0h, +4h, +10h, etc.), instead of fixed external clock time (7AM, 11AM, 5PM, etc.), which could have been in contradiction with the individual's normal sleep-wake timing. By designing sessions in function of the individual's timing of awakeness, sessions would have been in line with the sleep-wake pattern of each individual and still comparable between subjects, without the disadvantage of potentially impacting sleepiness and the underlying neurophysiology.

2. Furthermore, since one of the aims was to predict sleepiness, it would have been good practice to administer sleepiness (e.g. Epworth Sleepiness Scale (ESS)) and chronotype questionnaires in the recruitment phase. Ideally, it would have been also better to quantify the sleep-wake pattern of volunteers during the week before the study, ideally with actigraph or at least with sleep diaries.

Clarification and extension of the main conclusion (e.g. as reported in lines 92-94).

1. "While our results strongly support the synaptic homeostasis theory that synaptic connection strengthens overtime during wakefulness, […]". Authors should explain and reconcile the reduction in excitability during sleep deprivation (as inferred from MT and MEP latency) with the synaptic homeostasis theory (and other studies based on TEP).

2. "[…], we showed that the strengthening occurs in both excitatory and inhibitory synapses." Authors should clarify that this conclusion is based on a mixed-effect model and it is not supported by the raw data, which do not show any significant modulation of the excitatory and inhibitory profiles with time spent awake. Furthermore, it would be beneficial for the paper's clarity if the authors report the β values as well as their "R2", and eventually depict this with a graph. Finally, the authors reported "Pilot explorations adding sex and age as predictor variables show no effect of these variables." (line 352). It would be interesting to know if the predictor "session" was also tested in pilot explorations.

*Reviewer #3:*

Zhang and colleagues investigated the link between cortical excitability and self-reported sleepiness using a combination of resting-state electroencephalogram (EEG), non-invasive brain stimulation (TMS), and a sleep-deprivation intervention. Their key finding is that a subset of TMS measures including the amplitude and latency of the motor-evoked potential, plus measures of intra-cortical inhibition and facilitation, collectively predict a participants' degree of subjective sleepiness across the sleep-deprived period.

The underlying data for this paper is very strong. The authors have collected data from a good number of volunteers (N=38), and have acquired a broad range of TMS measurements; 6 different measurements acquired up to 8 times each. This richness has many advantages: It ensures good statistical power and enables the core mixed-model approach. It also allows the authors to link their results to deeper mechanisms, e.g. by including paired-pulse protocols that probe both cortical excitation and inhibition. That's really important for linking the results to models of sleepiness specified at the synaptic level, rather than just correlating TMS and sleepiness.

However some aspects of the paper require further attention. One is implementational and related to the model fitting. One is conceptual and relates to the behavioural self-reports.

1) The authors claim that the 4 TMS measures predict scores on the sleepiness scale. Therefore I was expecting some kind of cross-validation: partition data into subsets, obtain regression co-efficients from one subset, test the model fit on the other subset, repeat for many partitions. The authors do something a little less robust: they obtain regression co-efficients from 15 subjects and test on the other 23. Only partitioning once seems insufficient, and I couldn't find justifications for the choice of training on n = 15 testing on n = 23. It seems important to rule out that the results depend on the details of their fitting procedure, and to show that the model fit is truly robust.

2) The authors have demonstrated a link between TMS measures of cortical excitability and self-reported sleepiness. However, the link between self-reported sleepiness and synaptic potentiation seems weaker. The authors explain in the intro that learning during wakefulness leads to synaptic potentiation, and eventually circuit hyperactivity that is rebalanced by sleep. In that case, shouldn't longer wake time lead to more hyperactivity? In fact, in their data longer wake time does not lead to greater subjective sleepiness; after 24h of wakefulness the participants report feeling relatively wakeful again (Figure 1C). Is this prediction made by the synaptic homeostasis theory? If so, the paper would be stronger if the authors explained the prediction in the introduction. If not, then it's hard to see how results pertaining to subjective sleepiness support a conclusion about synaptic homeostasis.

[Editors' note: further revisions were suggested prior to acceptance, as described below.]

Thank you for resubmitting your work entitled "Cortical excitability signatures for the degree of sleepiness in human" for further consideration by *eLife*. Your revised article has been evaluated by John Huguenard (Senior Editor) and a Reviewing Editor.

The manuscript has been improved but there are some remaining issues that need to be addressed, as outlined below:

The authors made a deep and accurate revision, with some new control analyses. They responded to the points that were raised by the reviewers, and the manuscript is now quite improved. Two concerns need further consideration:

1. The authors provide some support to the relative independence from circadian factors. Please discuss the current results within the context of previous studies of cortical excitability as a function of circadian factors (e.g. Chellappa et al. Circadian dynamics in measures of cortical excitation and inhibition balance. Sci Rep. 2016 Sep 21;6:33661. doi: 10.1038/srep33661).

2. Concerning the theta and α activity across sleep onset, please also consider a preliminary finding on intracranial recordings of a pharmaco-resistant patient with epilepsy [supplementary Figure 1 in Marzano et al., 2013 (Sleep Med. 2013 Nov;14(11):1112-22. doi: 10.1016/j.sleep.2013.05.021)]

[Editors' note: further revisions were suggested prior to acceptance, as described below.]

Thank you for resubmitting your work entitled "Cortical excitability signatures for the degree of sleepiness in human" for further consideration by *eLife*. Your revised article has been evaluated by John Huguenard (Senior Editor) and a Reviewing Editor.

The manuscript has been improved but there are some remaining issues that need to be addressed. In particular, we note that there were apparently some significant changes in the author list, which is a bit unusual at this stage of the process, as the current revisions were relatively minor. Can you please provide a rationale for each of the authorship changes?

---

## [Author Response]

Essential revisions:1. The main concern raised by all three reviewers regards the impossibility to disentangle between the contributions of circadian vs. homeostatic factors on the observed effects. The analyses have been performed only considering the homeostatic factor (i.e., the time spent awake). We recommend that you address this concern by the following additional analysis: comparisons between data points collected at the same time-of-day (i.e., controlling the circadian factor): 3 data points for the TMS measures [7 a.m, 11 a.m., and 5 p.m. of the first and second day (before vs. after sleep deprivation)] and 1 data point for EEG measures (11 a.m. of the first and second day).

We thank the reviewers for pointing out circadian factors can be a confounding factor in our analyses. Following the suggestions from the reviewer, we now implemented PLS regression (similar to Figure 3A) on matched timepoints with TMS (new Supp. Figure 4A) and EEG data (new supp. Figure 3B) towards SSS. We found that for the 7 a.m. and 11 a.m. timepoints, both TMS and EEG data show obvious separation between sleepiness states, indicating that these measurements indeed represent a strong homeostatic component that is independent of circadian effect. We did not observe good separation in the 5 p.m. TMS group. We argue that this is due to the fact that within this timepoint the average sleepiness showed small difference.

In addition, to further test whether the variance in TMS measurements reflect homeostatic (sleepiness) factors or circadian factors, we grouped similar SSS datapoints and constructed mixed-effect linear models between TMS and hour of the day (new Supp. Figure 4B). In all groups, we did not observe correlation between TMS measurements and circadian factors, corroborating our conclusion that TMS measurements reflect homeostatic factors.

2. Linking the wakefulness measure to synaptic homeostasis theory may be possible via literature references. It would be important to explain (in the introduction, at least) what synaptic homeostasis theory predicts about a >24h sleep-deprivation intervention including the 'rebound' effect on sleepiness after 24h, since that is what the authors have done.

We have now included text introducing the circadian effect in the introduction section (line 61-63). Additional analyses related to circadian effects are now included in the results and Discussion sections.

3. It would also make sense to test the model more rigorously with something like 5-fold cross-validation.

We agree with the reviewers that cross-validation is a common practice for machine learning analysis. In general, however, there is no consensus on how to perform cross-validation for mixed-effect models due to the nature of random effects cannot be predicted for new subjects. It is an area of active research in the field of statistics regarding how to circumvent this issue. Currently, potential solutions including estimating information criteria of the model (*Statistical Science*, 2013, 28(2), 135–167; *Journal of Data Science*, 2011, 9, 15-21) or using an iterative method to estimate random effects (*Journal of Pharmacokinetics and Pharmacodynamics*, 2013, 40, 243–252).

We implemented the second method and used the prediction error as the metrics for measuring model accuracy. Among 38 subjects, we used 60% (23 subjects) for training and 40% (15 subjects) for testing. Within the training subjects, we used a 5-fold random sampling to choose 18 subjects for estimating the initial model, and 5 subjects for evaluating the error. This is repeated for 500 times and the best model is refitted using the 23 subjects for estimating the random effects and calculating the training error. Next, we applied this model to the 15 subjects testing set by freezing the fixed effect and estimating the random effects, and finally calculating the testing error. We repeat this training-testing process for 500 times to get a robust and generalizable result. As a negative control, we used randomly shuffled SSS dataset for the same procedure.

These cross-validation analyses are plotted in the new Figure 3E. The training and testing errors are close to each other, indicating robust predicting power from TMS measurement to SSS. On the other hand, shuffled data produced chance level estimation, further supporting the validity of the model.

4. Line 352: "Pilot explorations adding sex and age as predictor variables show no effect of these variables." It would be interesting to know if the predictor "session" was also tested.

We have tested “session” as a predictor as the reviewers requested. We coded session into numeral 1-8 in the model. The result showed that “session” showed significant correlation to SSS:

**Author response table 1. sa2table1:** 

Fixed effects coefficients (95% CIs):							
Name	Estimate	SE	tStat	DF	pValue	Lower	Upper
{'Session'}	0.31713	0.043097	7.3584	115	2.99E-11	0.23176	0.4025
{'MT' }	0.99275	0.30134	3.2944	115	0.001312	0.39584	1.5897
{'MEPL' }	1.2578	0.27701	4.5407	115	1.39E-05	0.70911	1.8065
{'ICF' }	0.099822	0.31145	0.32051	115	0.74916	-0.5171	0.71674
{'SICI' }	0.48192	0.30182	1.5967	115	0.11307	-0.11593	1.0798

However, we do not think “session” is a very meaningful variable as it contains mixed information about homeostatic and circadian information. As shown in Figure 1C, SSS is grossly correlated with session. Thus, this correlation is quite expected.

5. Figures 1.C. and 2.C-E.: if there are any significant differences between sessions, please report it in the figure.

We have now provided the pair-wise post-hoc tests for these panels in Supp. Figure 2.

6. Figure 1.D..: could you discuss the higher δ and theta activity observed in occipital areas (instead of fronto-central areas)?

We have now added this discussion in the result section (line 115-118). In summary, we observed consistent trends with data from individual EEG electrode but indeed see more prominent changes in the occipital regions. This is consistent with previous studies showing that sleep-associated changes happen most prominently in the occipital regions. This may reflect that our subjects are drifting in and out of sleep states throughout the program.

7. Line 128: MT abbreviation without previous definition in the text (but in line 307).

We have added the definition at its first appearance in the text (line 134).

8. Methods / Statistical analysis: specify if the main assumptions required by mixed-effect model were checked and met.

The core assumption of mixed-effect model is that the residuals and random effect coefficients are independent and identically distributed (*Methods in Ecology and Evolution* 11(9), 1141-1152, 2020). We have plotted these variables in our model:

We have added the description of this calculation in the method section (line 406-421), without including Author response image 1 , as we feel that this might be too technical for the readers.

**Author response image 1. sa2fig1:** 

9. Methods / Study flow: give a brief rational why SSS (instead of, for example, KSS, VAS, etc.).

We have added this information in the method section (line 310-311).

10. An interesting analysis could be to test if the model could also predict the EEG activity (mean or theta), as an objective marker of sleepiness (even though EEG was only recorded 3 times in a subgroup of 15 participants).

We thank the reviewers for the suggestion. We applied canonical correlation analysis between TMS and EEG measurements due to their multi-dimensional nature. We reported the results in the new Supp. Figure 5. There are 2 modes between TMS and EEG show significant correlations. The first mode primarily aligns with the sleepiness measurements, but not the second mode. This result indicates sleepiness is a major covariate that correlates between EEG and TMS.

11. The study participants do not actually report feeling very sleepy. The key behavioral measure is self-reported sleepiness on the Stanford Sleepiness Scale which is a 7-point scale, with only 6-7 indicating the participant is 'sleepy'. However participants reported level 6-7 sleepiness only at approximately 15 observations (fig3D, grey dots). This is from 38 participants * 8 testing moments i.e., >300 total observations. If participants were only actually sleepy in <5% of observations this could weaken the conclusions about sleep and sleepiness. And the clustering of the observations around the lower points 1-5 on the Sleepiness Scale might reduce confidence in the model fit. This is not necessarily a big issue but it might be sensible to acknowledge it in the discussion.

We agree with the reviewers that it would have been better if more level 6-7 datapoints are in the model. Nevertheless, our model does show a gradient effect that matched well for SSS levels 1-5. We have revised the manuscript to include this issue (line 278-282).

[Editors' note: further revisions were suggested prior to acceptance, as described below.]

The manuscript has been improved but there are some remaining issues that need to be addressed, as outlined below:The authors made a deep and accurate revision, with some new control analyses. They responded to the points that were raised by the reviewers, and the manuscript is now quite improved. Two concerns need further consideration:1. The authors provide some support to the relative independence from circadian factors. Please discuss the current results within the context of previous studies of cortical excitability as a function of circadian factors (e.g. Chellappa et al. Circadian dynamics in measures of cortical excitation and inhibition balance. Sci Rep. 2016 Sep 21;6:33661. doi: 10.1038/srep33661).2. Concerning the theta and α activity across sleep onset, please also consider a preliminary finding on intracranial recordings of a pharmaco-resistant patient with epilepsy [supplementary Figure 1 in Marzano et al., 2013 (Sleep Med. 2013 Nov;14(11):1112-22. doi: 10.1016/j.sleep.2013.05.021)]

In this revision, we have made the following changes of the manuscript:

1. We have expanded our discussion on the potential confounding effect of circadian rhythm (lines233-245).

2. Regarding the findings in Marzano et al., 2013, we added two related discussions:

a. On our results of the occipital lobe showing the most prominent power in EEG(lines 123-126);

b. On the potential regional differences among brain regions (lines 285-290).

3. We have made some miscellaneous changes regarding authorship, affiliations and acknowledgements.

[Editors' note: further revisions were suggested prior to acceptance, as described below.]

The manuscript has been improved but there are some remaining issues that need to be addressed. In particular, we note that there were apparently some significant changes in the author list, which is a bit unusual at this stage of the process, as the current revisions were relatively minor. Can you please provide a rationale for each of the authorship changes?

We apologize for the confusion caused by the change on the author list made during our last revision. Specifically, we moved Dr. Hua-Teng Cao from 10th to 4th on the author list.

Dr. Cao is a postdoc scholar in Dr. Zhang’s laboratory and has been joining the project since the beginning, assisting Dr. Zhang on the analysis of the dataset. During the first round of revision, Dr. Cao has devoted substantial effort in completing the additional analyses. All corresponding authors (Drs. Zhe Zhang, Ti-Fei Yuan and Rui-Ping Hu) agreed that he should be acknowledged for his contribution. In re-submitting our manuscript, we have already changed Dr. Cao’s authorship in the online system, but failed to update the author list on the manuscript PDF file. We have corrected this error during our second round of revision, leading to the issue described above. Nevertheless, if you check the author list on the online system, you will see that his position was changed during the first round of revision, which correctly reflected his contribution to the study.